# Near-visible integrated soliton microcombs with detectable repetition rates

Peng Liu [1,4], Qing-Xin Ji [1,4], Jin-Yu Liu[1,4], Jinhao Ge[1,4], Mingxiao Li [2], Joel Guo [2], Warren Jin[2,3], Maodong Gao [1], Yan Yu[1], Avi Feshali[3], Mario Paniccia[3], John E. Bowers [2] & Kerry J. Vahala [1] ✉

Integrated soliton microcombs benefit a wide range of conventional comb applications through their compactness and scalability. And applications such as optical clocks and biosensing have driven interest in their operation at wavelengths approaching the visible band. However, increasing normal dispersion and optical loss at shorter wavelengths make short pulse operation at low pumping power challenging, especially for detectable-rate microcombs. Here, low-pump-power, detectable-rate soliton microcombs are demonstrated from telecom to visible bands using ultra-low-loss silicon nitride waveguides. Wavelength-multiplexed operation spanning 2/3 octave is also demonstrated in a single device. The results fill a gap needed for realization of integrated self-referenced visible microcombs.

Soliton microcombs[1], a class of integrated optical frequency combs[2], are well-optimized in the telecom band for application in massively parallel coherent signal sources[3–5], frequency dividers for optically-referenced microwave oscillators[6–8] optical clocks[9,10], multipliers in frequency synthesizers[5], light sources in dual-comb spectroscopy[11,12] and ranging[13,14], and frequency calibration sources[15,16]. However, extending their operation into the visible and near-visible bands remains a persistent challenge, limiting applications such as atomic-referenced microwave generation, biological spectroscopy, and probing of atomic transitions (e.g., Rubidium at 780 nm, Strontium at 698 nm). The primary challenges are increasing normal dispersion and waveguide loss with decreasing wavelength in all photonic dielectrics.

Waveguide losses increase the turn-on power required for microcomb generation, often exceeding laboratory laser power. Simultaneously, soliton mode-locking requires anomalous waveguide dispersion. For the latter, control of dispersion by geometrical or mode-coupling methods[17] has made possible soliton microcomb operation at 780 and 1064 nm[18–22]. However, on account of increasing loss (reduced optical $Q$ factor) in integrated resonator platforms, only one of these demonstrations operated at a detectable repetition rate[18]. And that demonstration used a non-integrated silica wedge resonator[23]

to attain $Q$ factors high enough to overcome the required large optical mode volume associated with low repetition rate operation. Despite being essential for most comb applications[2], there are so far no integrated microcombs that operate in the visible or near-visible bands with detectable rates.

Recently, silicon nitride ($Si_3N_4$) waveguides offering $Q$ factors well above 100 million have been introduced[24,25], opening-up low power and high coherence operation across a range of devices. These ultra-low-loss (ULL) waveguides attain high-$Q$ by using thinner $Si_3N_4$ layers (typically 100 nm or less), which has the side effect of causing normal dispersion in their waveguides, even in the telecom band. Nonetheless, mode locking by pulse-pair formation in coupled rings has been shown to overcome the inherent normal dispersion for low turn-on power operation, even at detectable repetition rates[26]. A curious feature of pulse pair mode locking is that effective anomalous dispersion reoccurs with a spectral periodicity determined by the microresonator design. This recurrence has been observed over a limited wavelength range[26]. But if combined with the demonstrated low propagation losses in the ULL $Si_3N_4$ waveguides across the infrared to visible spectrum[27,28], an important question is whether the pulse-pair mode locking modality enables detectable-rate microcomb operation at short wavelengths.

[1]T. J. Watson Laboratory of Applied Physics, California Institute of Technology, Pasadena, CA, USA. [2]ECE Department, University of California Santa Barbara, Santa Barbara, CA, USA. [3]Anello Photonics, Santa Clara, CA, USA. [4]These authors contributed equally: Peng Liu, Qing-Xin Ji, Jin-Yu Liu, Jinhao Ge. ✉e-mail: vahala@caltech.edu

In this work, we demonstrate pulse pair microcombs operating at 1550, 1064, and 780 nm bands. The combs offer for the first time both low turn-on powers and detectable rate operation in an integrated microcomb. Moreover, anomalous dispersion is demonstrated at wavelengths as short as 682 and 532 nm. Multiplexing of soliton microcombs for bichromatic operation at 1064 and 1560 nm using a single device is also demonstrated. When combined with the demonstration of near-visible high-repetition-rate microcombs (THz rate) that are octave span[22,29,30], these results make possible integrated self-referenced near-visible microcombs.

## Results

### Visible to infrared dispersion engineering

As illustrated in Fig. 1a, pulse pair mode locking uses two coupled, racetrack-shaped rings with a slight length difference. Coupling creates two supermodes (symmetric and antisymmetric) with corresponding frequency bands. The maximum second-order dispersion (positive for anomalous dispersion) for the antisymmetric band is given by (Supplementary Note)[26,31],

$$D_2 = D_{2,o} + \frac{2\pi\epsilon^2 D_1}{\tan(g_{co}L_{co})} \tag{1}$$

where $D_{2,o} < 0$ is the intrinsic normal dispersion of the waveguide, $\epsilon = (L_B - L_A)/(L_A + L_B)$ is the length contrast between two rings (circumferences $L_A$ and $L_B$), and $g_{co}$ and $L_{co}$ are the coupling strength and coupling section length, respectively.

Besides anomalous dispersion, high resonator $Q$ factor is crucial for soliton microcomb generation, as the pump threshold scales inversely with the square of $Q$[32]. However, it is challenging to achieve high $Q$ in integrated photonic platforms at shorter optical wavelengths, where the Rayleigh scattering and material absorption losses are higher. To address these problems, the coupled rings are fabricated using ULL Si$_3$N$_4$ waveguides fabricated in a CMOS foundry[24]

(Methods). Figure 1b presents photomicrographs of the coupled rings. Two standard Si$_3$N$_4$ thicknesses are employed: 100 nm thickness for near-infrared band devices (1064 and 1550 nm), and 50 nm for the visible band devices (532, 682, and 780 nm). High intrinsic $Q$ factors are measured (Fig. 1c): 47, 28, 31, 9.7, and 1.4 million for 1550, 1064, 780, 682, and 532, respectively.

Anomalous dispersion spectral windows from 1550 nm to as short as 532 nm were designed using finite element method (FEM) simulation, and then measured using broadband-tunable external cavity diode lasers (ECDLs) at the wavelengths shown in Fig. 1d. In the plots, the integrated dispersion, defined as $D_{int}(\mu) = \omega_\mu - \omega_0 - \overline{D_1}\mu$, is plotted where $\omega_\mu$ is resonant frequency at relative mode number $\mu$ ($\mu$ is defined such that $\omega_0$ is the mode frequency for $\mu = 0$ at wavelength shown in the upper horizontal scale). Also, $\overline{D_1}/2\pi$ is the average FSR of the two coupled rings. In each panel of Fig. 1d, the two dispersion curves correspond to the two supermodes created by mode coupling, where the upper curve corresponds to the antisymmetric mode (Fig. 1c) and exhibits anomalous dispersion (convex curvature) as required for short pulse soliton generation. The measured resonances (dots) agree well with the design targets (solid curves) in all plots. It is also noted that broadband tuning of dispersion by the Moiré speed-up effect[31] was applied in the 780 nm device. For this measurement, integrated heaters were fabricated as described in ref.[31].

### Soliton microcombs at 1550, 1064, and 780 nm

Figure 2 shows the optical spectra of the soliton microcombs when pumped at the anomalous dispersion centers of the corresponding devices in Fig. 1d. The parametric oscillation thresholds at 1550, 1064, and 780 nm are below 10 mW (Supplementary Note Fig. S1b), compatible with commercially available distributed feedback (DFB) lasers for potential hybrid integration[31,33]. All spectra are slightly biased towards lower frequencies due to the mode distribution between the two coupled rings[26]. In the 1550 nm soliton spectrum, two dispersive waves (DWs) are

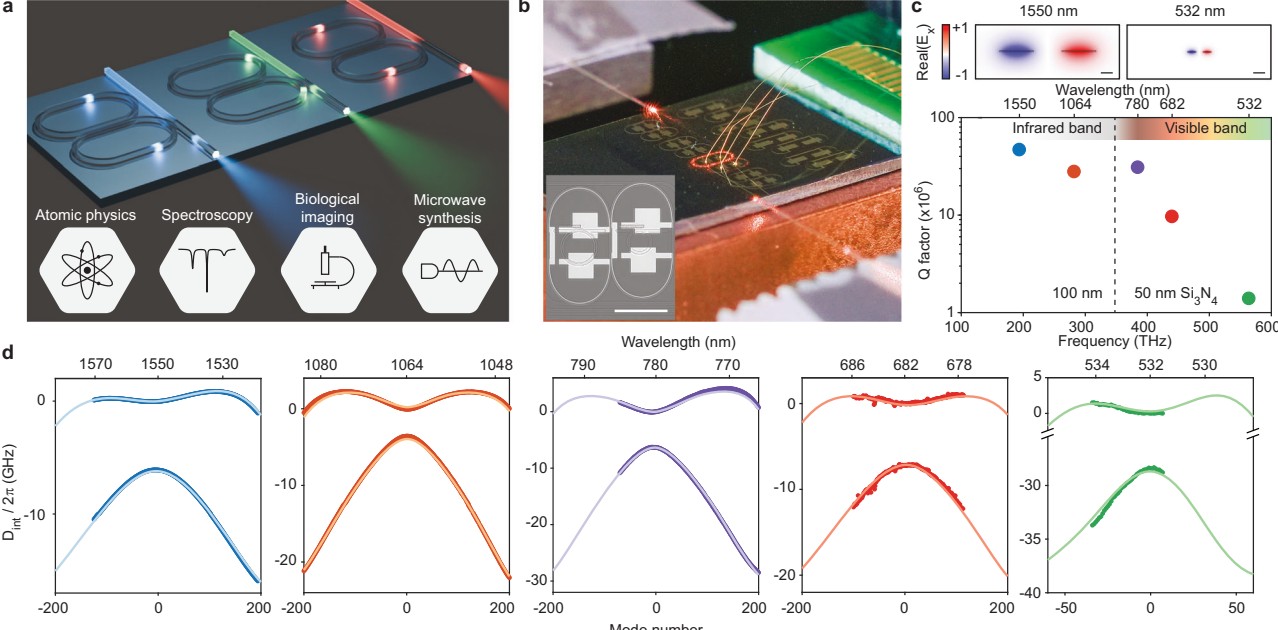

**Fig. 1 | Visible and near-visible anomalous dispersion in coupled-racetrack resonators. a** Conceptual illustration showing multiple, integrated, coupled resonators engineered for pulse-pair mode locking over a range of bands extending into the visible. **b** Photograph of coupled rings with 780 nm laser pumping. Integrated heaters are wire-bounded to an external printed circuit board for electrical dispersion tuning. Inset: photomicrograph of coupled rings with heaters deposited along the periphery. Scale bar: 1 mm. **c** Top panel: Cross-sectional view of simulated antisymmetric mode profiles at the coupling sections of 1550 and 532 nm coupled rings. Scale bars: 1 μm. Bottom panel: Measured intrinsic $Q$ factors of coupled rings from 1550 nm to 532 nm. **d** Dispersion measurements of five coupled racetrack resonators designed for 1550, 1064, 780, 682, and 532 nm. The solid curves are the design targets and are in good agreement with the measured dispersion (dots).

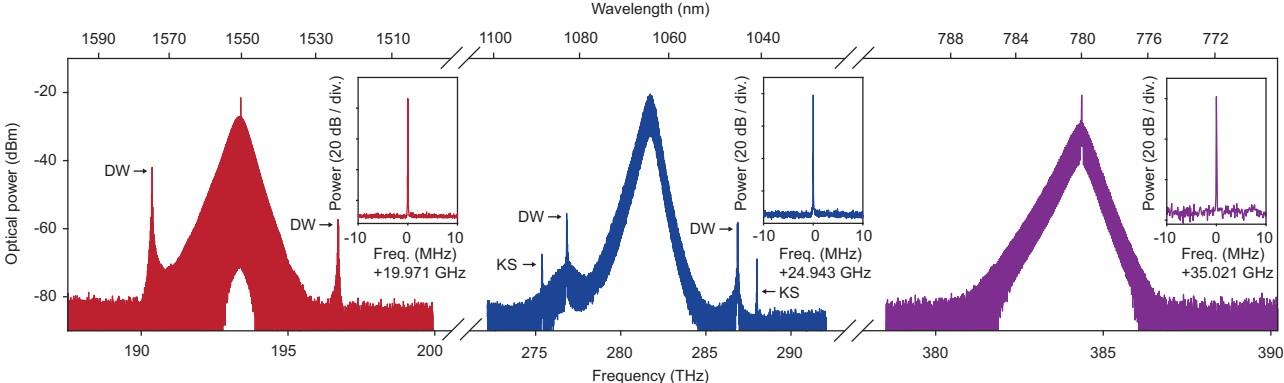

**Fig. 2 | Soliton microcomb spectral characterization.** Microcomb optical spectra are measured using devices designed for 1550, 1064, and 780 nm operation, respectively. In the 1064 nm soliton spectrum, the two spikes near 275.37 THz and 288.00 THz are interband Kelly sidebands (KSs). Dispersive waves (DWs) are also indicated in the 1550 and 1064 nm spectra. Insets: the electrical spectra of the photo-detected 1550, 1064, and 780 nm pulse streams. The resolution bandwidth in each spectrum is 1 kHz.

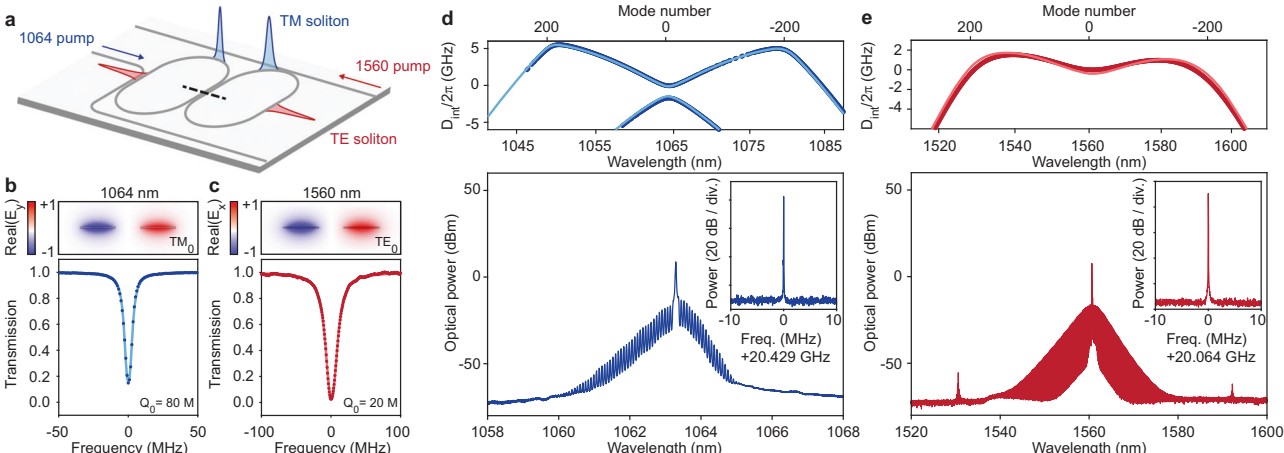

**Fig. 3 | Multiplexed solitons with 2/3 octave spacing. a** Illustration of soliton multiplexing on a standalone coupled racetrack device pumped at 1064 nm (TM mode) and 1560 nm (TE mode). **b, c** Upper panels: Simulated mode profile of the 1064 nm (1560 nm) antisymmetric $TM_0$ ($TE_0$) mode in the racetrack coupling section (dashed line in **a**). Lower panels: Transmission spectrum measurements of 1064 nm (TM) and 1560 nm (TE) mode resonances giving intrinsic $Q$ factor ($Q_0$) of 1064 (1560) nm of 80 million (20 million). **d, e** Upper panels: Integrated dispersion measurements of the coupled racetrack resonator near 1064 and 1560 nm bands. The solid curves give the design targets, and agree well with the measurements marked by dots. Lower panels: The optical spectra of the multiplexed solitons are shown in the corresponding lower panels. Insets: electrical spectra of the detected 1064 and 1560 nm soliton pulse streams. The resolution bandwidth is 1 kHz.

observed as expected for pulse pair mode locking[26]. Moreover, two interband Kelly sidebands (KSs) are present in the 1064 nm spectrum (Supplementary Note Fig. S2)[34]. Mode-locking of soliton microcombs is confirmed by photodetecting the soliton pulse stream and measuring the beatnote photocurrent with an electrical signal analyzer. The electrical spectra exhibit high-SNR single-tones, as shown in the insets of Fig. 2. Also, phase noise measurements of the beatnotes (Supplementary Note Fig. S3) indicate stable soliton mode locking. Supplementary Note Fig. S1a shows the experimental setups for the above measurements. Additional experimental details are provided in the Methods.

## 2/3 octave multiplexed microcombs

To achieve broadband coverage of integrated microcombs, several spectral extension techniques have been proposed and implemented based on multichromatic pumping[35–37]. Wavelength multiplexing makes possible operation on-demand across multiple spectral windows. As noted earlier, pulse-pair mode locking not only enables soliton microcomb generation within specific

wavelength ranges but also wavelength multiplexed operation. The pumping scheme is shown in Fig. 3a and uses orthogonally-polarized pumps for 1560 nm ($TE_0$) and 1064 nm ($TM_0$). Because the $TM_0$ mode has an overall larger mode area at a given wavelength compared to $TE_0$, this pumping scheme leads to mode balanced mode areas for TM and TE modes (see Fig. 3b, c) and correspondingly more balanced resonator loading at the shorter and longer wavelengths. Specifically, for a given mode, the coupling strength would otherwise tend to decrease at shorter wavelengths due to the stronger mode confinement in the waveguide (Supplementary Note Fig. S4).

Coupled rings with 1064 and 1560 nm bus-ring couplers were designed with corresponding antisymmetric mode profiles depicted in Fig. 3b, c. It is noteworthy that for the $Si_3N_4$ waveguide (3200 nm × 100 nm), the $TM_0$ resonance typically exhibits higher $Q$ factor than the $TE_0$ resonance at same wavelength due to reduced overlap with the waveguide[38,39]. Specifically, the intrinsic $Q$ factor ($Q_0$) is around 20 million for 1560 nm $TE_0$ and 80 million for 1064 nm $TM_0$ (as fitted by Lorentzian lineshapes of

the transmission in Fig. 3b, c). The higher $Q$ factor benefits comb generation at 1064 nm with lower power consumption (Supplementary Note Fig. S5).

Measured dispersion spectra (Fig. 3d, e) and calculated coupling dispersion (Supplementary Note Fig. S4) show that soliton generation can be supported by the $TE_0$ mode from approximately 1700 nm to 1400 nm and the $TM_0$ mode from 1400 nm to 1000 nm. To generate orthogonally-polarized solitons, 1064 nm and 1560 nm lasers are amplified by Ytterbium- and Erbium-doped fiber amplifiers and separately follow the frequency-sweeping approach (Methods). Measured 1064 nm and 1560 nm soliton microcomb spectra are shown in Fig. 3d, e, respectively. The photo-detected beatnotes of the multiplexed microcombs are shown in the insets, together with corresponding phase noise measurements (Supplementary Note Fig. S6), confirming stable mode locking.

## Discussion

An adaptive dispersion engineering approach was used to generate anomalous dispersion windows at arbitrary wavelengths, ranging from the infrared (1560 nm) to the green (532 nm). Prototyped on a CMOS-foundry low-loss $Si_3N_4$ platform, detectable-rate soliton microcombs at 1550, 1064, and 780 nm were demonstrated. Also, multiplexed microcombs at 1064 nm and 1560 nm in a single device were demonstrated. At present, the repetition rates of these combs are different, which is potentially useful for mid-IR spectroscopy[40,41]. In principle, it is possible to design a resonator structure that enables equal repetition rates for the two combs (see Supplementary Note Fig. S7). Moreover, by incorporating higher-order modes to maintain coupling strength, the multiplexed soliton microcombs are expected to span a full octave (Supplementary Note Fig. S8). Using this adaptive dispersion engineering approach in combination with thinner $Si_3N_4$ layers[27], TM resonances[38] or materials with lower loss at the shorter wavelengths[28,42], it should be possible to extend soliton generation to even shorter wavelengths.

The low turn-on power of the soliton microcombs demonstrated here is within the range of commercial DFB lasers for hybrid integration[31,33], or potentially for heterogeneous pump integration[43,44]. By operating at detectable repetition rates, the devices demonstrated here fill a critical gap required for development of self-referenced microcomb systems in the visible and near-visible bands for application in precision spectroscopy, bioimaging, and quantum technology.

## Methods

### Coupled rings design

The coupled rings are designed using FEM simulations, following three steps.

First, the width and thickness of the $Si_3N_4$ waveguides are determined. Here, 50 nm and 100 nm thicknesses are used for visible and infrared bands as these are two standard thicknesses in the CMOS foundry. A narrower waveguide supports dilute mode families but introduces additional scattering loss, and a thinner waveguide layer enhances the $Q$ factor at the expense of mode confinement.

Second, the coupling section between the two rings is designed based on Eq. (1). Given the design targets for $\overline{D_1}$ and $D_2$, the circumferences $L_A$ and $L_B$ of the two rings and the coupling phase $g_{co}L_{co}$ are calculated, with $0 < g_{co}L_{co} < \pi/2$ deliberately imposed to ensure optimal tolerance against fabrication variations. Notably, $D_2$ constrains the coupling phase while preserving one degree of freedom.

Third, bus-ring couplers are added and, based on the estimated intrinsic $Q$ factor, their coupling strength is designed to achieve critical coupling. The design of the coupled rings

presented in Fig. 3 follows the same procedure, with the $D_2$ values at 1064 nm and 1560 nm eliminating the degree of freedom between $g_{co}$ and $L_{co}$.

In this work, the FSRs of the coupled rings shown in Fig. 1d are designed to be 20, 25, 35, 25, and 65 GHz for wavelengths of 1550, 1064, 780, 680, and 532 nm, respectively. All these coupled rings are designed with $\epsilon = 1/400$, except the 532 nm coupled rings where $\epsilon = 1/100$.

### Dispersion measurements

Dispersion measurements use a method detailed elsewhere[18,45]. Briefly, a broadband-tunable ECDL is scanned with its frequency tracked by a calibrated Mach-Zehnder interferometer. By photo-detecting the device transmission and analyzing the mode-hop-free range of the ECDLs, the resonance frequencies are extracted. The ECDL used for measurements near 682 nm experienced occasional tuning discontinuities, resulting in the scattering of resonances observed in Fig. 1d.

Dispersion near 532 nm was measured using the 1064 nm ECDL in combination with frequency doubling using a periodically-poled lithium niobate (PPLN) module. Due to the limited PPLN bandwidth (approximately 40 GHz) at a given temperature, the dispersion of the 532 nm coupled rings is measured by tuning the 1064 nm ECDL in combination with varying the temperature of the PPLN module. Here, after fully utilizing the tuning range of the PPLN, segmented transmission spectra are collected and stitched together based on the resonance positions.

### Soliton stabilization

The soliton microcombs shown in Fig. 2 are generated by frequency-sweeping the pump laser from the blue-detuned to the red-detuned regime. For 1550 nm soliton microcombs, the ECDL is modulated by a QPSK module to generate a frequency-variable pump as a QPSK sideband. The amplified pump is coupled to the microcomb chip after polarization alignment, and the comb power is collected at the drop port of the coupled rings. The microcomb is stabilized by detection of the average comb power with servo control feedback to pump frequency[32,46]. For the 780 nm soliton microcomb, the pump is generated by frequency-doubling the 1560 nm tunable pump. For the 1064 nm soliton, frequency sweeping is achieved by modulating the piezo actuator of the 1064 nm ECDL. All three setups are summarized in Supplementary Note Fig. S1a.

### Parametric threshold measurements

The parametric oscillation threshold is investigated by recording the power of four-wave-mixing sidebands at different pumping powers. The pump laser is amplified to different power levels and swept from the blue-detuned to the red-detuned regime. The transmission is filtered by a fiber Bragg grating to collect the sideband power. The power of comb sidebands are plotted versus calibrated on-chip pump powers in Supplementary Note Fig. S1b, and the threshold is determined by linear fitting.

## Data availability

All data contributing to this study are published at figshare (https://doi.org/10.6084/m9.figshare.28826630.v1).

## Code availability

The analysis codes are published at figshare (https://doi.org/10.6084/m9.figshare.28826630.v1).

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

## Acknowledgements

The authors thank Myoung-Gyun Suh for loaning equipment, Selina Zhou, Shuman Sun, Doon Yoon Oh, Wei Zhang, and Hanfei Hou for helpful discussion. This work was supported by the National Science Foundation (Award Number 1908231) and the AFOSR (Award Number FA9550-23-1-0587).

## Author contributions

P.Liu, Q.-X.Ji, and K.J.Vahala conceived the idea. P.Liu, Q.-X.Ji, and J.Ge performed the measurement with assistance from M.Gao and Y.Yu. P.Liu, W.Jin, and A.Feshali designed the devices. P.Liu and J.-Y.Liu fabricated the device with the assistance from M.Li, J.Guo, A.Feshali, and M.Paniccia. All the authors analyzed the data and wrote the manuscript. J.E.Bowers and K.J.Vahala supervised the project.

## Competing interests

The authors declare no competing interests.
