## [Transparent Peer Review file · Nature Communications]

Near-visible integrated soliton microcombs with detectable repetition rates

Corresponding Author: Professor Kerry Vahala

Version 0:

Reviewer comments:

Reviewer #1

(Remarks to the Author)

The authors demonstrated soliton microcombs with detectable repetition rates within the visible and near-infrared wavelengths, utilizing an ultra-low-loss silicon nitride waveguide platform. They extended the soliton pulse pair scheme, originally designed to counteract the inherent normal dispersion of this on-chip integrable platform at 1.5 μm , to shorter wavelengths, successfully achieving anomalous dispersion at 682 nm and 532 nm. Leveraging this adjusted dispersion and high Q-factors, soliton mode locking was demonstrated at 1550, 1064, and 780 nm with input pump power under 10 mW. Notably, a single device exhibited multiple mixed microcombs at 1560 nm and 1064 nm using clever polarization multiplexing that accommodates similar mode profiles despite the large wavelength differences. According to the authors, these outcomes represent a crucial step towards creating a self-referenced microcomb system extended to the visible spectrum. The techniques developed in this work could also be applied to other on-chip material platforms to further broaden soliton generation to even shorter wavelengths. The experimental findings and their analysis are robust and well articulated. I support its publication in Nature Communications following revisions to address the subsequent points:

1. Although anomalous dispersion was shown, soliton generation wasn't achieved at 682 nm and 532 nm. Could the authors discuss potential methods for generating soliton microcombs at these wavelengths? Specifically, provide estimates of the required pump power with current quality factors, and identify the primary loss sources limiting Q-factors at these wavelengths with possible solutions to improve them.
2. The 2/3 octave multiplexed microcombs are compelling. For applications like locking with a high repetition rate broad-spectrum microcomb, these dual-color combs need individual control. Since they utilize two separate pumps, control can be achieved through pump detuning. Please elaborate on the control technique and discuss any potential interference, such as thermal interference, between the two pumps.
3. The supplementary materials offer phase noise measurements for microcomb beat notes at 1550 nm, 1064 nm, and 780 nm. Many readers would be interested in the phase noise characteristics of the 2/3 octave multiplexed microcombs. Please include these results in the manuscript or supplement and discuss any differences between the single soliton and multiplexed solitons.
4. Please include a discussion on extending the color separation of multiplexed microcombs from 2/3 octave to a full octave.

Reviewer #2

(Remarks to the Author)

In this paper, the authors demonstrated anomalous dispersion in coupled ring resonators in high-Q thin SiN platforms from near IR to visible wavelengths. Anomalous dispersion is of critical importance in generating soliton microcombs for a wide range of applications. Soliton microcomb generation is demonstrated in 780, 1060, and 1550 nm wavelength bands. All soliton microcombs have detectable repetition rates, and the threshold for parametric oscillation is all below 10 mW. Overall, this is a great demonstration of dispersion engineering with coupled ring resonators in visible wavelength, and its applications in soliton generation. The soliton microcombs in visible wavelength can be of critical importance to optical clocks, imaging and other comb-based applications. The paper is well-written, the data are clearly presented. I strongly recommend this paper for publication on Nature Communications.

Below are a few technical comments:

1. Could the authors comment on why the Q-factor drops significantly below 780 nm? I don't quite understand why this will be the case for thin-SiN. As most of the optical modes are in the oxide cladding layer, the surface roughness dependence on wavelength shouldn't have such a significant effect on loss.

2. On page three, first paragraph: "Thresholds for soliton microcomb generation at 1550 nm, 1064 nm, and 780 nm are below 10 mW (Fig. S1b)."

Figure S1b measured thresholds of parametric oscillation (or comb generation), instead of soliton generation. These two thresholds are different.

3. The 2/3 octave multiplexed microcombs generated from the same coupled ring resonators are quite interesting. I noticed that their comb repetition rate is off by roughly 0.4 GHz, probably due to dispersion. Can the authors comment on how to use this type of multiplexed microcombs given that they don't share the same repetition rate?

Version 1:

Reviewer comments:

Reviewer #1

(Remarks to the Author)

The authors thoroughly addressed all queries and requests, revising the manuscript accordingly. Their experimental results and accompanying analysis are both robust and well articulated. This work holds substantial impact, as it can extend soliton generation to shorter wavelengths across various on-chip material platforms with normal dispersion. Therefore, I strongly recommend its publication in Nature Communications.

Reviewer #2

(Remarks to the Author)

The authors have addressed all my questions. I recommend it for publication.

Dear Editor and Reviewers,

We sincerely appreciate the time and effort invested by the reviewers in evaluating our manuscript. Their insightful comments and suggestions have been invaluable in improving our work.

In the revised manuscript, we have added new experimental and simulation results to the Supplementary Note and refined the main text to better present the results. In the following, we present our point-by-point reply (in blue) to the reviewers' remarks (in black), as well as our actions taken (in red).

Reviewer #1 (Remarks to the Author):

The authors demonstrated soliton microcombs with detectable repetition rates within the visible and near-infrared wavelengths, utilizing an ultra-low-loss silicon nitride waveguide platform. They extended the soliton pulse pair scheme, originally designed to counteract the inherent normal dispersion of this on-chip integrable platform at 1.5 μm , to shorter wavelengths, successfully achieving anomalous dispersion at 682 nm and 532 nm. Leveraging this adjusted dispersion and high Q-factors, soliton mode locking was demonstrated at 1550, 1064, and 780 nm with input pump power under 10 mW. Notably, a single device exhibited multiple mixed microcombs at 1560 nm and 1064 nm using clever polarization multiplexing that accommodates similar mode profiles despite the large wavelength differences. According to the authors, these outcomes represent a crucial step towards creating a self-referenced microcomb system extended to the visible spectrum. The techniques developed in this work could also be applied to other on-chip material platforms to further broaden soliton generation to even shorter wavelengths. The experimental findings and their analysis are robust and well-articulated. I support its publication in Nature Communications following revisions to address the subsequent points:

We thank the reviewer for speaking highly of our work and supporting its publication.

We addressed all the comments as follows.

1. Although anomalous dispersion was shown, soliton generation wasn't achieved at 682 nm and 532 nm. Could the authors discuss potential methods for generating soliton microcombs at these wavelengths? Specifically, provide estimates of the required pump power with current quality factors, and identify the primary loss sources limiting Q-factors at these wavelengths with possible solutions to improve them.

Firstly, we estimate the parametric oscillation threshold at 532 nm as suggested by the reviewer.

As the platform for 532 nm coupled rings are the same as 780 nm coupled rings [on the same 50 nm silicon nitride (SiN) wafer], the parametric oscillation threshold can be calculated from parameters scaled from 780 nm. The parametric oscillation threshold is

$P_{th} = \frac{\pi n \omega_0 A_{eff}}{4 \eta n_2 D_1 Q_T^2}$, where n is the effective index, ω_0 is the pump frequency, A_{eff} is the effective mode area, $\eta = Q_T/Q_E$ is the loading parameter, D_1 is the free spectral range, and $Q_{T/E}$ are the loaded / external quality factors (Q's).

We listed the estimation parameters in the table below and assumed that n_2 is energy-weighted average of silica and SiN, where $n_{2,SiN} \sim 11 \times n_{2,silica}$ (inferred from 1550 nm values in Ref. [1]). To our knowledge, frequency dependence in visible band of Kerr parameter n_2 in thin-film LPCVD SiN is not presented anywhere. Therefore, we simply suppose that $n_2(532 \text{ nm}) \simeq n_2(780 \text{ nm})$ for both SiN and silica.

Parameters	532 nm	780 nm
n	1.53	1.49
ω_0	$2\pi \times 564 \text{ THz}$	$2\pi \times 385 \text{ THz}$
A_{eff}	$0.23 \mu\text{m}^2$	$1.30 \mu\text{m}^2$
Q_E	5.8 M	84 M
Q_i	1.4 M	31 M
Q_T	1.1 M	23 M
n_2	1.4	1 (norm.)
D_1	$2\pi \times 65 \text{ GHz}$	$2\pi \times 35 \text{ GHz}$
P_{th}	619 mW (est.)	9.58 mW (mea.)

The estimated pump threshold for oscillation is $> 600 \text{ mW}$ for the current device, which is not achievable with laboratory lasers near 532 nm. Experimentally, we managed to deliver 20 mW to the 532 nm coupled rings, but there is no nonlinear phenomenon detected.

Secondly, we believe that the significant Q drop in visible band is due to the scattering loss, which arises from the roughness of both the top and bottom surfaces and the waveguide sidewalls. To do estimations, we used data in Ref. [2], where the fabrication process is similar with the CMOS foundry that provided samples to us. The data used for scattering simulation is $\sigma B = 50 \text{ nm}^2$ for sidewalls and $\sigma B = 2 \text{ nm}^2$ for the top and bottom surfaces. The calculated scattering Q is normalized to 532 nm measured Q and listed in Fig. R1b, which agrees reasonably well with the measurement at 680 nm and

780 nm. This calculation identifies that the surface roughness of the waveguides is the primary loss source.

Thirdly, to further enhance the Q-factors at 532 nm band, a potential method is to use SiN waveguides with wider width or thinner thickness. Also, utilizing TM resonances can minimize the overlap with waveguide sidewalls that is rougher than the top and bottom surfaces, which potentially benefits Q-factors at 532 nm band such as in Ref. [3-4].

2. The 2/3 octave multiplexed microcombs are compelling. For applications like locking with a high repetition rate broad-spectrum microcomb, these dual-color combs need individual control. Since they utilize two separate pumps, control can be achieved through pump detuning. Please elaborate on the control technique and discuss any potential interference, such as thermal interference, between the two pumps.

The on-chip powers for wavelength-multiplexed solitons at 1560 nm and 1064 nm are respectively around 500 mW and 80 mW, where high Q of TM mode benefits the power consumption at 1064 nm. The thermal interference is therefore weak as the pumping power of 1064 nm is much lower than 1560 nm.

To lock the wavelength-multiplexed solitons to a high repetition rate broad-spectrum microcomb, individual detuning of two pump lasers near 1550 nm and 1064 nm can be applied for separated control and locking.

Moreover, on-chip metallic heaters can be deposited along two racetrack resonators as presented in Ref. [5]. Two metallic heaters provide two degree-of-freedom to manipulate 1064 nm and 1560 nm properties such as pump mode frequencies. With differential tuning of temperature and different thermo-optic coefficients between 1064 nm and 1560 nm modes, comb lines can be reconfigured to match external frequency lines [6].

3. The supplementary materials offer phase noise measurements for microcomb beat notes at 1550 nm, 1064 nm, and 780 nm. Many readers would be interested in the phase noise characteristics of the 2/3 octave multiplexed microcombs. Please include these results in the manuscript or supplement and discuss any differences between the single soliton and multiplexed solitons.

We thank the reviewer for their interest in noise performance of the multiplexed solitons. The phase noises of multiplexed solitons at 1560 nm and 1064 nm are measured and included in the Supplementary Note Fig. S6. The comparison between multiplexed solitons and single solitons are as follows:

1. Comparing the single soliton and multiplexed solitons at 1560 nm, we found that the phase noise from 1 kHz to 1 MHz frequency offset are closely aligned.
2. Comparing the single soliton and multiplexed solitons at 1064 nm, we found that the phase noise from 1 kHz to 1 MHz frequency offset are aligned if taking $\left(\frac{20.429 \text{ GHz}}{24.943 \text{ GHz}}\right)^2$ FSR scaling factor into consideration.
3. For 10 Hz to 1 kHz frequency offset, the phase noises deviate some moderate amount because of the servo locking bump and low frequency technical noise.

We added the following to the main text:

The photo-detected beatnotes of the multiplexed microcombs are shown in the insets, together with corresponding phase noise measurements (Supplementary Note Fig. S6), confirming stable mode locking.

4. Please include a discussion on extending the color separation of multiplexed microcombs from 2/3 octave to a full octave.

We thank the reviewer for completing the manuscript with further extending the color separation of multiplexed soliton microcombs to full octave. The pulse-pair mode-locking method is extendable to 780 nm band by using higher order TM mode (TM₃), which emphasize its transferability. We included the discussion and simulation to the Supplementary Note Fig. S8 and added the following to the main text:

Moreover, by incorporating higher-order modes to maintain coupling strength, the multiplexed soliton microcombs are expected to span a full octave (Supplementary Note Fig. S8).

Reviewer #2 (Remarks to the Author):

In this paper, the authors demonstrated anomalous dispersion in coupled ring resonators in high-Q thin SiN platforms from near IR to visible wavelengths. Anomalous dispersion is of critical importance in generating soliton microcombs for a wide range of applications. Soliton microcomb generation is demonstrated in 780, 1060, and 1550 nm wavelength bands. All soliton microcombs have detectable repetition rates, and the threshold for parametric oscillation is all below 10 mW. Overall, this is a great demonstration of dispersion engineering with coupled ring resonators in visible wavelength, and its applications in soliton generation. The soliton microcombs in visible wavelength can be of critical importance to optical clocks, imaging and other comb-based applications. The paper is well-written, the data are clearly presented. I strongly recommend this paper for publication on Nature Communications.

We thank the reviewer for speaking highly of our work and strongly recommending its publication. We addressed all the comments as follows.

Below are a few technical comments:

1. Could the authors comment on why the Q-factor drops significantly below 780 nm? I don't quite understand why this will be the case for thin-SiN. As most of the optical modes are in the oxide cladding layer, the surface roughness dependence on wavelength shouldn't have such a significant effect on loss.

We thank the reviewer for the interest in loss performance of our thin-film SiN platform. Indeed, our Q drop in visible band is more significant than the sidewall surface roughness limit (Q versus wavelength is usually $\sim \lambda^3$ as presented in Ref. [7-9]). We believe that the significant Q drop in visible band is due to the roughness of both the top and bottom surfaces and the waveguide sidewalls, as the cross section of 780 nm single-mode coupled rings are different from the cross section of 532 nm single-mode coupled rings.

Firstly, we want to exclude the absorption limit. For absorption limit estimation, the cross-section of waveguides, mode profiles, and power composition are summarized in

Fig. R1a. We expect absorption $Q > 20$ million for 532 nm and longer wavelength. This is because in Ref. [7-8], the absorption Q for 532 nm is about 6 million for high confinement SiN waveguides. For our thin-film SiN waveguide, about 23% of power distributes in SiN waveguide core. So, we believe the 1.4 million 532 nm Q is not due to the absorption limit.

As suggested in Ref. [10-11], the scattering loss can be modeled by dipole radiations on the waveguide surfaces. Neglecting far-field interference, the dipole radiation power can be simply estimated as

$$P_{scat} \propto \frac{1}{\lambda^4} \times (n_{clad}^2 - n_{core}^2) \times \int (\sigma B)^2 E^2 dl$$

Where λ is the optical wavelength, $n_{clad/core}$ are the refractive indices of waveguide cladding / core, σ is the sidewall roughness RMS, B is the correlation length of roughness, and integration is along the boundary (top and bottom surfaces plus two sidewalls). The corresponding Q -factor scales as

$$Q_{scat} \propto \frac{\lambda^3 \times \int \epsilon_r E^2 dA}{(n_{clad}^2 - n_{core}^2) \times \int (\sigma B)^2 E^2 dl}$$

where scattering is normalized by energy flow in cavity.

To do estimations, we do not have the roughness information of our foundry-processed SiN waveguides. However, we can use data in Ref. [2] for an estimate, where the fabrication process is similar. The data is $\sigma B = 50 \text{ nm}^2$ for sidewalls and $\sigma B = 2 \text{ nm}^2$ for the top and bottom surfaces. The calculated scattering Q is normalized to 532 nm measured Q and listed in Fig. R1b, which agrees reasonably well with the measurement at 680 nm and 780 nm.

2. On page three, first paragraph: "Thresholds for soliton microcomb generation at 1550 nm, 1064 nm, and 780 nm are below 10 mW (Fig. S1b)." Figure S1b measured thresholds of parametric oscillation (or comb generation), instead of soliton generation. These two thresholds are different.

We thank the reviewer for providing this rigorous view. We changed this sentence to:

The parametric oscillation thresholds at 1550 nm, 1064 nm, and 780 nm are below 10 mW (Supplementary Note Fig. S1b), compatible with commercially available distributed feedback (DFB) lasers for potential hybrid integration.

3. The 2/3 octave multiplexed microcombs generated from the same coupled ring resonators are quite interesting. I noticed that their comb repetition rate is off by roughly 0.4 GHz, probably due to dispersion. Can the authors comment on how to use this type of multiplexed microcombs given that they don't share the same repetition rate?

We thank the reviewer for commenting this 2/3 octave multiplexed microcombs quite interesting. The comb repetition rate difference is indeed induced by the dispersion and local FSR difference. However, slightly different repetition rates are preferred in some case. For example, Ref. [12] and Ref. [13] have introduced the interleaved difference frequency generation (iDFG) using two frequency combs at 1.55 μm and 1.06 μm with different frequency repetition rates. The iDFG not only down converts the microcomb frequency to the mid-IR band as precise spectroscopy source, which is appealing for most gas species, but also increase the resolution of dual comb spectroscopy resulting from two different repetition rates. In our case, with two frequency combs generated in one resonator with different repetition rates (400 MHz difference), we can lock the repetition rate ratio to a rational ratio (i.e., 56/55 here). In this case we could generate a phase-locked mid-IR comb with around 400 MHz comb line spacing, providing high resolution in gas spectroscopy.

Moreover, by tailoring the geometry, thin-film Si₃N₄ can also support FSR-matched 2/3 octave multiplexed microcombs. We included a simulation of group index in the **Supplementary Note Fig. S7** and added following sentences in the main text:

Also, multiplexed microcombs at 1064 nm and 1560 nm in a single device were demonstrated, which is potentially useful for Mid-IR spectroscopy and spectral translated locking across 2/3 octave (Supplementary Note Fig. S7).

The simulation suggested that using commercially available SiN waveguide of 185 nm thickness and 1518 nm width enables group index matching, which means FSR-matching can be achieved across 2/3 octave by polarization multiplexing method.

Figures:

Fig. R1: Mode composition and scattering limit of visible Q factors. a Comparison of mode profile between 532 nm, 680 nm, and 780 nm. **b** Calculated scattering limited Q factors, which agrees reasonably well with the measurement.

Reference:

- [1]. Gao, Maodong, et al. "Probing material absorption and optical nonlinearity of integrated photonic materials." *Nature communications* 13.1 (2022): 3323.
- [2]. Puckett, Matthew W., et al. "422 Million intrinsic quality factor planar integrated all-waveguide resonator with sub-MHz linewidth." *Nature communications* 12.1 (2021): 934.
- [3]. Liu, Kaikai, et al. "Ultralow 0.034 dB/m loss wafer-scale integrated photonics realizing 720 million Q and 380 μ W threshold Brillouin lasing." *Optics letters* 47.7 (2022): 1855-1858.
- [4]. Morin, Theodore J., et al. "CMOS-foundry-based blue and violet photonics." *Optica* 8.5 (2021): 755-756.
- [5]. Ji, Qing-Xin, et al. "Multimodality integrated microresonators using the Moiré speedup effect." *Science* 383.6687 (2024): 1080-1083.
- [6]. Ji, Qing-Xin, et al. "Dispersive-wave-agile optical frequency division." *CLEO: Science and Innovations*. Optica Publishing Group, 2024.
- [7]. Corato-Zanarella, Mateus, et al. "Widely tunable and narrow-linewidth chip-scale lasers from near-ultraviolet to near-infrared wavelengths." *Nature Photonics* 17.2 (2023): 157-164.
- [8]. Corato-Zanarella, Mateus, et al. "Absorption and scattering limits of silicon nitride integrated photonics in the visible spectrum." *Optics Express* 32.4 (2024): 5718-5728.
- [9]. Wu, Lue, et al. "Hydroxyl ion absorption in on-chip high-Q resonators." *Optics Letters* 48.13 (2023): 3511-3514.
- [10]. Gorodetsky, Michael L., Andrew D. Pryamikov, and Vladimir S. Ilchenko. "Rayleigh scattering in high-Q microspheres." *Journal of the Optical Society of America B* 17.6 (2000): 1051-1057.
- [11]. Gorodetsky, Mikhail L., Anatoly A. Savchenkov, and Vladimir S. Ilchenko. "Ultimate Q of optical microsphere resonators." *Optics letters* 21.7 (1996): 453-455.

[12]. Bao, Chengying, et al. "Interleaved difference-frequency generation for microcomb spectral densification in the mid-infrared." *Optica* 7.4 (2020): 309-315.

[13]. Bao, Chengying, et al. "Architecture for microcomb-based GHz-mid-infrared dual-comb spectroscopy." *Nature communications* 12.1 (2021): 6573.